# Growth Factor Delivery Using a Collagen Membrane for Bone Tissue Regeneration

**DOI:** 10.3390/biom13050809

**Published:** 2023-05-10

**Authors:** Tadahiro Takayama, Kentaro Imamura, Seiichi Yamano

**Affiliations:** 1Department of Periodontology, Nihon University School of Dentistry, Tokyo 101-8310, Japan; 2Division of Advanced Dental Treatment, Dental Research Center, Nihon University School of Dentistry, Tokyo 101-8310, Japan; 3Department of Periodontology, Tokyo Dental College, Tokyo 101-0061, Japan; 4Oral Health Science Center, Tokyo Dental College, Tokyo 101-0061, Japan; 5Department of Prosthodontics, New York University College of Dentistry, New York, NY 10010, USA

**Keywords:** bone regeneration, growth factors, collagen membranes, osteogenesis

## Abstract

The use of biomaterials and bioactive agents has shown promise in bone defect repair, leading to the development of strategies for bone regeneration. Various artificial membranes, especially collagen membranes (CMs) that are widely used for periodontal therapy and provide an extracellular matrix-simulating environment, play a significant role in promoting bone regeneration. In addition, numerous growth factors (GFs) have been used as clinical applications in regenerative therapy. However, it has been established that the unregulated administration of these factors may not work to their full regenerative potential and could also trigger unfavorable side effects. The utilization of these factors in clinical settings is still restricted due to the lack of effective delivery systems and biomaterial carriers. Hence, considering the efficiency of bone regeneration, both spaces maintained using CMs and GFs can synergistically create successful outcomes in bone tissue engineering. Therefore, recent studies have demonstrated a significant interest in the potential of combining CMs and GFs to effectively promote bone repair. This approach holds great promise and has become a focal point in our research. The purpose of this review is to highlight the role of CMs containing GFs in the regeneration of bone tissue, and to discuss their use in preclinical animal models of regeneration. Additionally, the review addresses potential concerns and suggests future research directions for growth factor therapy in the field of regenerative science.

## 1. Introduction

There has been a significant increase in the number of oral rehabilitation surgeries involving the placement of dental implants and periodontal regeneration, which often requires the use of artificial membranes for guided bone regeneration (GBR) [1] and guided tissue regeneration (GTR) [2]. Clinical and/or surgical interventions are a mandatory step in the recovery of health of the periodontal tissues. To date, various regenerative surgical modalities have been suggested and examined for the regeneration of periodontal tissues [3]. These procedures have involved the use of diverse surgical techniques, barrier membranes, various bone grafts, and other materials that are osteoconductive/inductive, or mixtures of proteins and exogenous growth factors (GFs) [4].

Among these backgrounds, the use of osseointegrated implants for the rehabilitation of edentulous areas has significantly advanced dentistry and enhanced the quality of life of patients [5]. However, insufficient bone remains a major challenge for bone-anchored implants. The reconstruction of alveolar bone for appropriate dental implant placement is most frequently achieved through GBR, which is the most commonly used treatment approach. Additionally, this procedure generally refers to alveolar ridge augmentation or bone regenerative procedures. The procedure induces cell differentiation, cell proliferation, and the induction and/or conduction of tissue formation [1].

A promising strategy for restoring the tissue structure and function involves recreating the microenvironment and activating signal transduction pathways that promote the tissue regeneration at the site of injury [6]. The replication of essential regenerative pathways that facilitate tissue regeneration can be achieved by utilizing a combination of biomaterials, cells, and GFs. This approach promotes the functional and structural replacement of the tissue [7]. GFs are crucial signaling molecules that are secreted by cells and play a vital role in various cellular processes necessary for healing, including proliferation, migration, and differentiation. Moreover, they are polypeptides that regulate cellular growth, differentiation, and metabolism. These GFs also have a significant impact on wound healing and repair processes [8]. Against this background, we consider that utilizing GFs is a promising approach to achieve optimal tissue regeneration.

The significance of delivery systems in the successful application of GFs will be examined, followed by an overview of biomaterial substrates and techniques utilized for their delivery. Notably, all clinically approved GF therapies employ some form of a biomaterial carrier for the delivery of GFs [9]. This highlights the crucial role that biomaterial delivery systems play in the successful application of GFs in regenerative medicine. In recent years, autologous platelet concentrates (APCs), characterized by abundant GF levels, exhibit remarkable suitability for periodontal and bone tissue regeneration procedures. The main generations of APCs are platelet-rich plasma [10], platelet-rich fibrin [11], and concentrated growth factor [12] techniques. The advantage of the APCs is the fact that a large amount of GF can be delivered to the target site, and they encourage angiogenesis and wound healing. However, these regenerative approaches do not always yield satisfactory results due to various limitations, including complex surgical procedures, chronic inflammation, and insufficient clinical predictability. Therefore, we have been considering a novel and efficient delivery system for GFs and exploring the possibility of a new treatment for severe bone defects using a combination of GFs and collagen membranes (CMs).

Collagen-based materials, such as membranes, sponges, matrices, hydrogels, and composite scaffolds, are extensively utilized in in vivo studies to facilitate the regeneration of bone tissue [13]. The collagen sponge is highly regarded as a valuable biomaterial due to its exceptional functionality and properties, along with its convenient processing, sterilization, and storage capabilities [14]. Recently, Briquez et al. [15] demonstrated that a collagen sponge combined with bone morphogenetic protein (BMP)-2 and a bridge protein enhanced the safety of BMP-2 and its beneficial impact on bone regeneration for spinal fusion. An effective strategy for bone regeneration was observed by incorporating a gene delivery system containing the fibroblast growth factor (FGF-2) and BMP-2 within a collagen sponge, resulting in optimal bone regeneration effects [16]. Additionally, in the rabbit model of full-thickness bone defect regeneration, the utilization of a hydrogel collagen nanocomposite in conjunction with strontium and seeded mesenchymal stem cells improved bone formation [17]. The collagen/hydroxyapatite (HA) composite biomaterial scaffolds have been extensively investigated and used for bone tissue engineering scaffolds [18]. HA-collagen is very effective in healing critical-sized bone damage in a rodent model [19].

Herein, we provide a critical overview that particularly focus on reviewing the efficacy for collagen barrier membranes with GFs of our previous studies. We have developed new applications using CMs containing GFs that have both functions, preventing the migration of soft tissue growth to maintain spaces for bone and accelerating bone healing in an effort to further enhance clinical outcomes, including the speed, quality, and quantity of newly formed bone. In addition, we have searched for new candidate GFs for an effective method of bone regeneration. As in the first part of the review, the characteristic of CMs as a scaffold will be thoroughly addressed. Thereafter, we discuss the GFs related to osteogenesis that we have investigated and the effect of CMs containing GFs for GBR is summarized and evaluated based on the results of our in vitro and in vivo research reported. Last, the author’s opinions in this review may serve as a foundation for further interdisciplinary opinions to obtain optimized CMs with GFs for GBR application and provide future directions and challenges for this strategy.

## 2. CMs

Barrier membranes have been widely employed in dentistry to supplement bone augmentation in implant therapy and periodontal regenerative dentistry. GBR and GTR have been extensively researched and are now widely accepted as a fundamental procedure for regenerating periodontal tissue loss [20]. The conventional method involves placing a barrier membrane on a regenerative site where volumetric tissue loss has occurred, with the aim of preventing the migration of undesired cells from the gingival epithelium and connective tissue [2]. As a result, the barrier membrane creates a space that facilitates tissue regeneration, allowing for the selective guidance of periodontal ligament cells and bone regeneration towards the desired outcome [21]. In order to achieve good clinical regenerative outcomes, it is important for barrier membranes to possess certain ideal properties. These include biocompatibility and non-toxicity towards the surrounding tissue and the body, as well as a high tissue tolerance to promote the complete integration with the periodontal fibers. Additionally, adequate structural integrity and dimensional stability are necessary to ensure that the membrane maintains its shape and position until degradation or removal, while still allowing for tolerable clinical handling, suitable storage time, and easy modification and application. The membrane should also be selectively permeable to prevent epithelial cell invasion while promoting osteogenic cell proliferation, as well as being capable of maintaining space for regenerative cells and promoting adequate blood clot formation for angiogenesis and vascularity during regeneration.

Type I collagen is the most abundant type of collagen in the extracellular matrix (ECM) of mammals, and it is found predominantly in tissues such as periodontal connective tissue, tendon, and bone [22]. The ECM is crucial for the morphogenesis and cellular metabolism of new tissues, providing mechanical and biochemical properties [23]. Collagen has been found to be a useful biomaterial for bone tissue engineering because of its abundance, biocompatibility, high porosity, ability to combine with other materials, easy processing, hydrophilicity, low antigenicity, and ability to be absorbed by the body [24]. The nanotopographical feature of collagen type I fibers directly stimulate the process of osteogenic differentiation in both human and mouse mesenchymal stem cells as well as neural crest-derived stem cells, without requiring additional biochemical signals for bone formation [25]. Moreover, it is widely recognized that nanotopography of artificial materials play a crucial role in determining the fate of stem cells [26], influencing gene expression, and promoting the attachment of bone-forming cells [27]. This nanopotography is shown to be solely sufficient for guiding osteogenic differentiation of osteogenic related cells in vitro [28]. In addition, the transplantation of a collagen sponge (Spongostan) with a nantopographical structure into a critical-size calvarial rat bone defect further leads to the fast regeneration of the lesion in vivo [29]. Therefore, it is suggested that the nanotopographical structure in the collagen membrane used in our study also played a crucial role in bone formation.

Natural resorbable membranes, known as CMs, are typically sourced from human, porcine or bovine tissues such as the pericardium, dermis, and Achilles’ tendon [30]. CMs possess appealing characteristics such as their compatibility with biological systems, ability to control bleeding, ability to attract cells to a wound site, and capacity to promote wound healing by stabilizing blood clots [31]. CMs have been found to be effective in promoting bone regeneration due to their ability to inhibit the growth of epithelial cells and their low immunogenicity [32]. Although CMs are useful for bone regeneration, their regenerative capacity can be limited by early resorption. To overcome this issue, several crosslinking methods have been developed to extend their absorption time and improve periodontal regeneration. These techniques involve the use of substances such as glutaraldehyde (GA), formaldehyde, ultraviolet light, hexamethylenediisocyanate, and diphenylphosphorylazide. Without crosslinking, CMs can be resorbed within a range of 5 to 28 days [33]; crosslinked membranes, on the other hand, can maintain their structural integrity for up to 14 days, which is enough time to prevent epithelial downgrowth during the early stages of wound healing. Furthermore, a higher degree of crosslinking in collagen membranes results in slower resorption rates. However, the prolonged presence of the membrane may not be conducive to optimal regeneration during wound healing. In a double-blinded randomized controlled trial, it was observed that highly crosslinked collagen has an adverse effect on regeneration and prolongs the resorption rate in cases of membrane exposure [34]. Highly crosslinked CMs exhibit a significantly higher incidence of tissue dehiscence compared to native CMs, indicating that a membrane barrier may not be necessary beyond the initial stage of wound healing.

The use of CMs for bone defect regeneration in oral implant surgery has become prevalent due to their bioresorbable nature, which eliminates the need for a secondary surgical procedure [35]. To date, resorbable membranes that are commercially available have been utilized for the treatment of periodontal and peri-implant defects through GTR, which is based on the principle of epithelial exclusion [36]. The use of resorbable CMs in GTR and GBR procedures has become widespread due to their well-established biocompatibility and ability to facilitate wound healing. Recently, collagen materials have also been applied in GBR with comparable success rates compared to non-absorbable expanded polytetrafluoroethylene membranes. The exceptional biocompatibility and cell affinity of CMs have resulted in favorable regenerative outcomes [37]. Although CMs demonstrated excellent biocompatibility and cell affinity for tissue regeneration, the mechanical strength of the membranes was found to be insufficient. Additionally, the degradation rate of CMs did not align with the natural tissue-healing process; novel approaches have been devised to enhance both mechanical properties and biocompatibility of degraded membranes. Various collagen cross-linking methods have been employed to increase membrane biodurability and prolong membrane reabsorption, as previously discussed.

A typical commercially available CM products used in routine clinical practice is BioMend^®^ (BM; Zimmer Dental, Carlsbad, CA, USA), which is composed of glutaraldehyde cross-linked bovine type I collagen fibers. Collagen fibers make up the majority of the bone matrix, and serve as a storage site for numerous local factors. Additionally, they play a crucial role in connecting osteogenic cells to the surrounding matrix [38]. Although GBR membranes lack proteins specific to bone, their collagen fibers can still act as a structural framework for osteogenic cells in bone defects. Moreover, they provide a protective barrier that prevents the penetration of adjacent connective tissues. Native collagen degrades in a few days, untreated CMs lack stability, and space is not available in the absence of bone support. Various crosslinking techniques have been developed to solve these problems. Crosslinking is the proliferation of spontaneous bonds between collagen molecules. This increases the strength of the collagen membrane and delays enzymatic degradation. These properties appear to be dose- or time-dependent. The number of collagen crosslinks is also proportional to both the membrane stiffness and degradation time. The most frequently used crosslinking method is the treatment of collagen with GA. GA has been demonstrated to increase the cytotoxicity of CMs in vitro [39] and in vivo [40]. GA is known to be released during collagen degradation. OsseoGuard^®^ (OG; Collagen Matrix Inc/Biomet 3i in Palm Beach Gardens, FL, USA) is a bovine-derived collagen type I membrane. This membrane has a pore structure that effectively blocks the invasion of gingival and epithelial cells, while allowing the passage of vital nutrients and gases. In a series of our studies to date, we have used BM and OG, which are natural collagen-based artificial membranes. Each membrane has been demonstrated to have the moderate membrane absorption rates and to promote a favorable healing process. In addition, we look for an absorbable membrane that is suitable for bone regeneration units while investigating the toxicity of each membrane to tissues. It is necessary to have characteristics of biocompatibility, an absence of biological side effects, an efficient release of GFs, and biodegradability as the desired criteria for our study.

## 3. GFs for Osteogenesis

Up to date, many GFs involved in bone regeneration have been studied and discovered. The most common growth factor known to promote bone formation is the bone morphogenetic protein (BMP)-2, which has been approved by the Food and Drug Administration (FDA) for clinical use. However, it still has many problems such as clinical side effects, difficulty in controlling the concentration used, and cost–benefit issues. Therefore, we have been searching for GFs that are safer for living organisms and more efficient for osteogenesis, and have conducted a series of in vitro and in vivo studies. The following is a summary of the characteristics of the representative GFs candidates we have investigated.

### 3.1. Bone Morphogenetic Protein (BMP)-2

BMP-2, a recombinant bone morphogenetic protein, is the growth factor most frequently used to stimulate bone formation in humans. It has been approved by the FDA for its safety and effectiveness. BMP-2 is known to enhance the recruitment and angiogenesis of osteoblast progenitor cells and has garnered significant attention for its exceptional osteoinductive capabilities. However, the commercial application of BMP-2 is faced with two major challenges. The first is its instability and rapid degradation in vivo, which necessitates the loading of high doses into scaffolds. This can result in exorbitant costs and an increased risk of side effects such as inflammatory reactions [41], nerve injury [42], heterotopic ossification [43], and tumor development [44]. The second challenge is that absorbable collagen sponges [45] and calcium phosphates [46] are currently the only two carriers approved for BMP-2 loading in clinical treatments. However, the current carriers approved for BMP-2 loading, such as absorbable collagen sponges and calcium phosphates, have a poor affinity for BMP-2. This results in a serious risk of burst release and low retention rates, which forces the use of higher doses and exacerbates local and systemic adverse effects. To address these problems and enhance the bone healing process, various strategies have been developed to improve BMP-2 delivery while preserving its bioactivity. These strategies include immobilization approaches and the engineering of high retention scaffolds [47,48]. Further research on BMP-2 is necessary to evaluate its long-term effects and assess the need for alternative biologics or scaffolds. Additionally, a cost–benefit analysis of BMP-2 in the healthcare system should also be performed.

### 3.2. Platelet-Derived Growth Factor (PDGF)

PDGF, a two-chain polypeptide, is synthesized by osteocytes, platelets, and mono-cyte-macrophage cells and occurs in several isoforms, including PDGF-AA, PDGF-AB, PDGF-BB, PDGF-CC, and PDGF-DD. PDGF-BB is a powerful cytokine that interacts with all three PDGF receptors: PDGFR-αα, PDGFR-αβ, and PDGFR-ββ [49]. One important function of this regulator is to control the process of tissue repair and reconstruction, including the promotion of bone healing and the facilitation of angiogenesis [50]. PDGF is released by platelet aggregation at the site of bone repair and then diffuses into the tissue to act as a chemoattractant, thereby migrating cells involved in tissue repair. This activates undifferentiated mesenchymal cells, which are then transformed into osteoblasts. This results in accelerated bone growth and is one of the main factors in cellular infiltration into the site of bone repair. Additionally, PDGF is known to stimulate the chemotaxis and growth of osteoblasts during the process of bone healing [51], as well as improve bone regeneration in individuals who have difficulty healing due to systemic conditions such as osteoporosis and diabetes [52]. The recombinant human PDGF-BB isoform (rhPDGF-BB) is approved by the U.S. Food and Drug Administration (FDA) for clinical dentoalveolar and peri-implant bone regeneration [53]. Recent research has demonstrated that PDGF can increase the regeneration of both soft and hard tissues in the periodontal area, as well as improve the healing of bone in extraction sockets. Furthermore, PDGF has the ability to enhance the performance of MSCs, which can accelerate the regeneration of bone in the maxillofacial region [54].

### 3.3. Growth/Differentiation Factor-5 (GDF-5)

GDF-5, also known as cartilage-derived morphogenetic protein-1, is a member of the BMP (bone morphogenetic protein) superfamily. GDF-5 shares 40–50% of the protein sequence homology with BMP-2 and BMP-7 [55]. Studies suggest that GDF-5 affects angiogenesis, migration, apoptosis, and differentiation in vitro [56]. The presence of GDF-5 is essential for the differentiation of mesenchymal cells and the development of tendon, skeletal, and ligament tissues [57]. GDF-5 is capable of promoting the growth of human periodontal ligament cells. It belongs to the BMP family of dimeric proteins, which are involved in the formation of bone and cartilage. BMPs have various regulatory functions in different cell types, including the control of cell growth, differentiation, migration, and apoptosis [58]. The role of GDF-5 is crucial in the differentiation of mesenchymal cells, as well as the development of skeletal, tendon, and ligament tissues [59]. The local application of GDF-5 promotes bone formation in rat calvarial defects [60] and enhances long bone healing [61]. Moreover, GDF-5 promotes the periodontal ligament cell proliferation affecting ECM metabolism [62]. In vitro studies have demonstrated that GDF-5 can substantially enhance the growth of primary osteoblasts, periosteum cells, and connective tissue fibroblasts [63]. In vivo studies have demonstrated that the combination of GDF-5 and β-TCP has been proven effective in promoting bone growth through bone regeneration procedures. This approach has been tested successfully in both sinus floor augmentation [64] and the treatment of bone defects around dental implants located in the mandible [65].

### 3.4. Stromal Cell-Derived Factor 1 (SDF-1)

SDF-1, better known as chemokines, are in the dynamics of circulating hematopoietic and mesenchymal stem cells. They have also been reported to contribute to angiogenesis and bone regeneration [66]. In addition, SDF-1 is induced in the periosteum proximal to the damaged bone and migrates MSCs to that site, promoting endochondral ossification [67]. Previous studies have demonstrated that this growth factor is involved in angiogenesis and directly affects bone formation [68]. Overexpression of SDF-1 has also been proven to promote bone regeneration by inducing angiogenesis [68]. It has also been used in a study by Guobao Chen et al. involving segmental bone defects in the rabbit radius, concluding that the application of SDF promotes bone formation [69]. SDF-1 is also involved in the recruitment of inflammatory cells and undifferentiated mesenchymal cells committed to other types of tissues [70]. Recently, SDF-1 has been demonstrated to play an important role in the mobilization and recruitment of circulating bone marrow-derived hematopoietic stem cells and mesenchymal stem cells (MSC), contributing to cell-mediated angiogenesis and bone regeneration. In addition, SDF-1 has been found to be required for blood homeostasis [71] and bone remodeling [72]. Recent reports indicate that SDF-1 also recruits human periodontal ligament stem cells in vitro [73]. SDF-1 has also been demonstrated to be involved in the recruitment of human bone marrow-derived hematopoietic stem cells and MSCs.

### 3.5. Osteogenic Protein-1 (OP-1)

OP-1, also known as osteogenic protein-7 (BMP-7), has been extensively analyzed biochemically since its gene was identified in the late 1980s, acts on both bone and cartilage in vitro [74]. Previous studies have demonstrated that the effect of OP-1 on cell proliferation depends on the cell type and culture conditions. Furthermore, OP-1 has been found to stimulate the production of both alkaline phosphatase (ALP) and osteocalcin in osteoblast-like MC3T3-E1 cells in a dose-dependent manner [75]. In addition, OP-1 not only promoted the maturation of osteoblastic progenitor cells, but also induced the differentiation of undifferentiated mesenchymal cells into osteoblasts [76]. In summary, OP-1 was demonstrated to induce osteogenesis. As for its effect on intracellular signaling, when OP-1 binds to cell surface receptors, Smad1 and 5 are phosphorylated and complexed with Smad4 to activate Runx2, a key transcription factor that controls osteoblast differentiation, maturation, and ECM [77]. Its osteoinductive properties have led to the use of OP-1 as a therapeutic agent for the treatment of bone fractures and as an osteogenic agent at bone graft sites. Various animal models have been used to evaluate the therapeutic effects of OP-1 in bone repair. Many in vivo studies have used OP-1 in combination with biocompatible and/or biodegradable polymers and collagenous carriers and scaffolds on the basis that the osteoinductive capacity of OP-1 can promote bone defect regeneration [78]. In dental practice, hydroxyapatite (HA)-coated rhOP-1 has been found to promote the healing of bone defect sites after tooth extraction and promote the osseointegration of dental implants with high initial stability. rhOP-1 application to bone defect sites has been demonstrated to promote bone healing with a greater volume, density, and degree of bone remodeling [79]. The combination of rhOP-1 with a collagen carrier derived from bovine has been found to have the capability to stimulate the bone formation in the human maxillary sinus following sinus floor elevation [80].

### 3.6. Fibroblast Growth Factor-18 (FGF-18)

FGF-18 belongs to the FGF family and is expressed primarily in the lungs, kidneys, and several organs [81,82]. FGF-18 is known to play an important role in bone growth and development [83]. FGF-18 is also known to be an important regulator of bone growth and differentiation. FGF-18 is a multifaceted growth factor that promotes the proliferation and differentiation in a variety of tissues [84] and is an essential autocrine positive regulator of the osteogenic differentiation program in mouse mesenchymal stem cells [85]. Ossification is delayed in mice lacking FGF-18 [86]. Furthermore, FGF-18 has also been thought to have the ability to improve bone repair and regeneration [87]. Long bones from FGF18-/- mice had reduced the osteoblast differentiation potential. These results indicate that FGF-18 positively affects osteogenesis by promoting cell proliferation and differentiation [83]. An in vitro analysis using bone marrow-derived mesenchymal stem cells (MSCs) suggested that FGF-18 promotes osteoblast differentiation by activating FGFR1 or FGFR2 signaling [85]. Treatment of rat-derived MSCs with FGF-18 under differentiation-inducing conditions resulted in the increased expression of osteoblast differentiation markers and mineralization [87].

### 3.7. Fibroblast Growth Factor-2 (FGF-2)

FGF-2 is the most common growth factor used in regenerative medicine, including bone regeneration; FGF-2 is a known growth factor that induces neovascularization and osteogenesis; members of the FGF family are 16–18 kDa monomeric proteins with a variety of biological functions, including cell migration, cell differentiation, angiogenesis, wound healing, and osteogenesis, among various other biological functions [88]. FGF-2 also promotes osteoblast proliferation and is a potent mitogen for MSCs. FGF-2 is well known to be an important component in the maintenance of undifferentiated mesenchymal cells [89]. In FGF-2-/- mice, a marked decrease in bone mass and bone formation was observed. FGF-2-/- mice bone marrow stromal cells (BMSCs) demonstrated reduced osteoblast differentiation, while the addition of exogenous FGF-2 in vitro partially restored their function [90]. Furthermore, FGF-2-/- BMSC-derived osteoblasts demonstrated a marked decrease in Dkk2 mRNA, which plays an important role in osteoblast differentiation. These results suggested that FGF-2 acts endogenously and positively as a regulator of bone mass [91]. The addition of FGF-2 to the culture medium of the mouse osteoblast-like cell line MC3T3-E1 activated cell proliferation and inhibited mineralization [92]. In addition, randomized clinical trials have indicated that FGF-2 can effectively enhance the growth of alveolar bone in patients suffering from chronic periodontitis [93], and other studies have demonstrated that the local application of recombinant FGF-2 directly to sites of bone defects using a suitable carrier has been demonstrated to stimulate bone growth [94].

## 4. A Growth Factor Released from a CM

Sustained release kinetics is preferred when an extended time of GFs released is needed, especially in bone regeneration. It helps to deliver the drug at a predetermined rate, thereby extending the duration of sustained release and thereby maintaining a constant drug supply [95]. This kinetics is adapted when the lifespan of the drug in body fluids is very short and the excretion rate from the body is high. In the case of drug delivery for bone regeneration, this type of kinetics is desirable because it promotes a prolonged duration for the development of new bone. In the CMs with GFs unit developed in our study, this release kinetics has been confirmed for up to 14~21 days [96,97,98,99,100,101]. The total cumulative amount of sustained release ranged from 30% to ≈100%.

We have been using the aforementioned GFs, in combination with CMs both in vitro and in vivo to devise a bone regeneration unit that is useful for bone formation. The main focus of the in vitro studies was on the effect on osteoblasts, while the in vivo studies were on the improvement of bone loss in a rat mandibular bone defect model. The following is a summary of the results and discussion of each study (Table 1).

### 4.1. CMs with PDGF (CM/PDGF) [101]

The purpose of this study was to examine the ability of CMs as a carrier to efficiently deliver PDGF and to subsequently observe its effects on preosteoblasts in vitro. MC3T3-E1 mouse preosteoblasts were cultured in a CM containing PDGF. After a 2-day cell-culture, cell viability was evaluated by a 3-[4,5-dimethylthiazol-2-yl]-2,5-diphenyltetrazolium bromide (MTT) assay and cell proliferation was assessed by a crystal violet proliferation assay. Expression levels of osteoblast differentiation marker genes, a runt-related transcription factor 2 (RUNX2), osteopontin (OPN), bone sialoprotein (BSP), and osteocalcin (OCN) were measured by real-time polymerase chain reaction (RT-PCR). A sustained release of PDGF from CMs was observed for about 3 weeks. All gene expression of RUNX2, OPN, BSP, and OCN was significantly promoted in CMs with PDGF compared to without PDGF. Furthermore, PDGF released from CM was observed to significantly elevate the alkaline phosphatase (ALP) activity and cell proliferation with little cytotoxicity. These results suggest that CM can be useful for the sustained release of PDGF. The delivery of PDGF can also facilitate MC3T3-E1 cell activity. The implementation of this approach has the potential to enhance the current clinical management of bone deficiencies in regenerative therapy.

### 4.2. CMs with GDF5 (CM/GDF-5) [96]

The aim of this study was to compare the effects of PDGF and GDF-5 released from CMs on bone healing in vitro and in vivo. In vitro studies were performed using MC3T3-E1 mouse preosteoblasts cultured with or without these factors. Cell viability, cell proliferation, and ALP activity were measured. In in vivo studies, CMs supplemented with low or high doses of PDGF or GDF-5 were placed in the mandibular defects of rats. In in vitro studies, CM/GDF-5 markedly elevated the ALP activity and cell proliferation in MC3T3-E1 cells without cytotoxicity compared to CM/PDGF or CM alone. The gene expression demonstrated that Runx2 and OCN were considerably promoted by CM/GDF-5 compared to CM/PDGF or control. Quantitative and qualitative micro-computed tomography (μCT) and the histological analysis of osteogenesis revealed that CM/PDGF and CM/GDF-5 substantially accelerated bone regeneration compared to CM alone or controls. GDF-5 was also demonstrated to induce the new bone formation in a dose-dependent manner. These results suggest that this strategy for the application of CM/GDF-5 may improve the current clinical management of bone lesions.

### 4.3. CMs with SDF-1 (CM/SDF-1) [97]

The purpose of this study was to evaluate the effect of SDF-1 released from CM on bone regeneration compared to PDGF in vitro and in vivo. For in vitro studies, MC3T3-E1 mouse preosteoblasts were cultured in CM containing these factors, then cell proliferation, the ALP activity, and osteoblast differentiation marker genes were evaluated. A rat mandibular bone defect model was used to investigate the in vivo effects of CM/SDF-1 or CM/PDGF. At 4 weeks postoperatively, the measurement of new bone formation was performed using μCT and histological examination. The results of in vitro studies demonstrated that CM/SDF-1 significantly increased cell proliferation, ALP activity, and gene expression of all osteogenic markers compared to CM alone or control, similar to CM/PDGF. The evaluation of the quantity and quality of newly formed bone using μCT revealed that the administration of SDF-1 resulted in a remarkable enhancement and stimulation of bone regeneration, in comparison to the controls and the group treated with CM alone. The promotion of bone formation in the SDF-1 group was dose-dependent and with levels similar to those measured with PDGF. These results suggest that the combination of CM and SDF-1 could potentially serve as a favorable option for delivering growth factors, replacing PDGF, in various clinical interventions that involve bone regeneration.

### 4.4. CMs with OP-1 (CM/OP-1) [98]

The purpose of this study was to investigate the effect of CM/OP-1 on a rat mandibular bone defect model. Eight rats operated on both sides of the mandible were used to create a total of 16 defects and divided into 4 groups. Group 1: no treatment (control), Group 2: CM alone, Group 3: CM containing a low dose of 0.5 μg of OP-1 (L-OP-1), and Group 4: CM containing a high dose of 2.0 μg of OP-1 (H-OP-1). At 8 weeks postoperatively, newly formed bone was evaluated by the μCT and histological analysis. A quantitative and qualitative analysis by μCT evaluated the amount of new bone formation, bone mineral density, and percentage of new bone area. At 8 weeks after the surgery, the utilization of CM/OP-1 resulted in a considerable and notable augmentation and stimulation of bone volume, bone mineral density, and the proportion of newly formed bone area, when compared to both the controls and CM alone. The improvements in bone regeneration observed in the groups treated with OP-1 were directly proportional to the dose administered. The utilization of CM/OP-1 has the potential to serve as a powerful osteoinductive agent and a favorable combination for promoting bone regeneration. The use of this composite device in osteogenesis could lead to more secure and predictable outcomes for bone regeneration in the future.

### 4.5. CMs with FGF-18 (CM/FGF-18) [99]

We have created a novel delivery system based on proprietary growth factor with CMs to induce bone regeneration. We hypothesized that FGF-18, a multifaceted protein that elevates proliferation in multiple tissues, could be a good candidate for bone reconstruction using our delivery system. We cultured mouse preosteoblasts with CM/FGF-18, CM/PDGF, and CM alone, and assessed the marker gene expression of osteoblastic differentiation, cell viability, cell proliferation, alkaline phosphatase activity, and mineralization. In addition, preosteoblasts were cultured with CM/FGF-18 or CM alone, and expression of microRNAs, particularly miR-133a and miR-135a, which are involved in suppressing osteogenic factors, were measured. The sustained release of FGF-18 from CM was followed over 21 days. CM/FGF-18 was found to be more effective than CM concerning the gene expression of type I collagen (Col-I), Runx2, OCN, Smad5, and osteopontin (OPN) was markedly upregulated in CM/FGF-18 compared to CM alone, and similar to that of CM/PDGF. In addition, CM/FGF-18 downregulated the expression of miR-133a and miR-135a. These results suggest that FGF-18 released from CM promotes osteoblast activity via the downregulation of miR-133a and miR-135a.

### 4.6. CMs with FGF-2 (CM/FGF-2) [100]

The aim of this research is to examine and analyze the impact of CM/FGF-2 on the healing process in a rat mandibular angular bone defect model. Critical-sized circular bone defects (4.0 mm diameter) were formed on both sides of the rat mandible. The defects were randomized to control, CM alone, or CM containing low (0.5 g) or high (2.0 g) concentrations of FGF-2. To determine the volume of newly formed bone, bone mineral density, and the percentage of the area where new bone was formed, real-time in vivo μCT were conducted at baseline, 2, 4, and 6 weeks. At 6 weeks, the mandibular specimens were evaluated histologically and histomorphometrically to evaluate the area of new bone regeneration. Real-time evaluation demonstrated that the volume, bone mineral density, and closure rate of the new bone formation area were notably greater in the CM/FGF-2 group than in the control and CM alone groups. In the high dose FGF-2 group, the volume and bone mineral density of the new bone formation were considerably greater at 6 weeks than at baseline. Histological evaluation confirmed the presence of osteoblasts, osteocytes, and blood vessels within the new bone formation. Real-time in vivo experiments indicated that CM/FGF-2 effectively enhanced bone healing within a critical-sized mandibular defect in rats, and new bone formation was detected early postoperatively.

## 5. Conclusions and Future Outlook

The application of CM/GF for the purpose of encouraging the regeneration of bone tissues has introduced a new frontier for addressing oral and maxillofacial conditions and will gain potential for more opportunities for clinical applications in the future. CM/GF is a novel therapy with a unique strategy offering a unified procedure and protocol for bone tissue regeneration. Using CM/GF can also introduce new perspectives to regenerative therapies in a new era in oral health reconstruction. Additional research is needed to investigate the possibilities of the combined therapy, in order to potentially incorporate it as a standard form of treatment in the future. In summary, taking into consideration the current data from pre-clinical studies, it is suggested that a clinical application of CM/GF promotes hard tissue healing and contributes to improving patients’ quality of life and early recovery of oral functions.

The primary objectives of regenerative therapies are to restore tissues that have been lost as a result of injury or trauma. The clinical application of CM/GF has a potential to create significant expectations in dentistry. However, their use in clinics still has many challenges for clinical application, and the potential of growth factor therapy in the field of regenerative medicine has still not been fully realized. Moreover, the current preclinical approaches achieved limited success in rat mandibular bone defects. Nonetheless, the emergence of bone tissue engineering holds the potential to address the deficiencies of current clinical approaches for treating bone defects. Considerable advancements have been achieved in refining the selection of scaffold materials and the fundamental aspects of manipulating growth factors. Over the past few years, numerous controlled release systems utilizing membrane scaffolds, including our studies, have been effectively developed to manage the delivery of growth factors for the purpose of enhancing bone regeneration. Although the growth factor delivery system utilizing membranes appears to be a logical and attractive approach, research efforts are being directed towards developing delivery systems that enable accurate localization and reduced dosage in order to address concerns regarding safety and cost-effectiveness. There are multiple considerations that must be recognized to develop successful CM/GF therapies. Additional research is necessary to gain a more precise understanding of how growth factors interact with the targeted tissue. This understanding would facilitate the identification of the role for growth factors in tissue regeneration and enable the development of delivery systems that maintain the presence of the growth factor at the necessary stages of regeneration. Moreover, comprehending the regenerative biology specific to the bone being treated would assist in determining the potential effectiveness of the selected growth factor. Additionally, it is also essential to conduct research on the interaction between the host tissues and the growth factors utilized in delivery systems. All of the GFs used in our experiments have been demonstrated to be involved in bone regeneration, and since these GFs have beneficial biological activities, they may act synergistically in achieving bone tissue regeneration. Conversely, it is important to ensure that membrane materials can be safely broken down and eliminated from the body once the desired outcomes have been accomplished. Additionally, consideration should be given to the potential toxicity of the degraded products of the material to the host tissue. Second, there are issues surrounding the safety, cost-effectiveness, and efficacy of growth factor utilization in a clinical setting that must be resolved. Hence, it is crucial to establish strong pre-clinical models that detail the safety and pharmacokinetics of the utilized growth factor. Moreover, because of the high cost of growth factor usage accompanied by the limited efficacy resulting from topical application, their use in clinical practice will require more time. This has been partially addressed by the development of delivery systems which work better than simple topical application. However, while delivery systems have been developed to improve growth factor efficacy, they still necessitate a substantial amount of growth factors to achieve a noteworthy clinical benefit because of the suboptimal loading and release at the target tissue site. Therefore, in addition to investigating the efficient loading of GFs to delivery systems, research should also focus on developing cost-effective methods for producing recombinant GFs. Although biomaterial carriers increase the availability of GFs, they do not ensure that the growth factor will remain stable for extended periods of use. One possible issue with biomaterial carriers is that they may not provide sufficient stability for the GFs over long periods of time. The GFs may be degraded by proteases produced by the host tissue before they can take effect. Therefore, it may be worthwhile to investigate the development of synthetic growth factor mimics that have similar activity to the GFs but are not subject to degradation by the host cells. Last, it is increasingly becoming evident that the utilization of a solitary growth factor may not be adequate to attain the most favorable tissue regeneration. Even in the natural healing procedure, multiple GFs are generated at different stages of regeneration. As a result, the trend of using a combination of GFs is becoming progressively popular, as it is realized that using a single GF may not be sufficient for optimal tissue regeneration. However, caution must be exercised in the order in which the GFs are used, which should mimic the natural regeneration process, as using all of them simultaneously may lead to minimal or undesirable effects. Therefore, it is necessary to focus on the development of advanced delivery systems that permit the gradual and sequential release of GFs during tissue regeneration, and this should be a key area of research for future growth factor delivery. The promising clinical outcomes of growth factor therapy in regenerative medicine require an interdisciplinary approach that involves various fields such as biomaterial science, engineering, chemistry, biology, pathology, and reconstructive surgery. This approach is crucial to develop novel therapies that can effectively treat complex wounds and pathologies and promote tissue regeneration. The future benefits of such therapies for patients are significant.

## Figures and Tables

**Table 1 biomolecules-13-00809-t001:** Summary of several preclinical studies using different growth factors with a collagen membrane for bone regeneration.

Study	Materials		Design		Reference
	Growth Factor	Collagen Membrane	In Vitro	In Vivo	
Yamano et al. (2011)	PDGF	OG	(1) Cell culture (MC3T3-E1) (2) Release kinetics (3) MTT assay (4) ALP activity (5) Cell proliferation (6) Real-time RT-PCR	N/A	[101]
Yamano et al. (2014)	PDGF GDF-5	OG	(1) Cell culture (MC3T3-E1) (2) MTT assay (3) ALP activity (4) Cell proliferation (5) Real-time RT-PCR	Rat mandibular bone defect model (1) μCT (2) Histological analysis	[96]
Takayama et al. (2017)	SDF-1	OG	(1) Cell culture (MC3T3-E1) (2) Release kinetics (3) MTT assay (4) ALP activity (5) Cell proliferation (6) Real-time RT-PCR	Rat mandibular bone defect model (1) μCT (2) Histological analysis	[97]
Ozaki et al. (2017)	OP-1	BM	(1) Cell culture (MC3T3-E1) (2) Release kinetics (3) MTT assay	Rat mandibular bone defect model (1) μCT (2) Histological analysis	[98]
Imamura et al. (2018)	FGF-18 PDGF	OG	(1) Cell culture (MC3T3-E1) (2) Release kinetics (3) MTT assay (4) ALP activity (5) Cell proliferation (6) Real-time RT-PCR (7) Mineralization assay	N/A	[99]
Furuhata et al. (2021)	FGF-2	BM	(1) Release kinetics	Rat mandibular bone defect model (1) μCT (2) Histological analysis	[100]

OG: OsseoGuard^®^; BM: BioMend^®^.

## Data Availability

Not applicable.

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
