# Peer review of "Growth Factor Delivery Using a Collagen Membrane for Bone Tissue Regeneration"

_biomolecules, 2023, doi:10.3390/biom13050809_

Round 1

Reviewer 1 Report

The review discusses a very relevant area of regenerative science that is growing interest. The review reported the role of collagene membranes and several growth factor in bone regenerative medicine. The first part of the manuscript is well structured and discussed. Then the authors examine the use of CM with GF mentioning only their studies. However the authors themselves declare that “numerous controlled release systems utilizing membrane scaffolds including our studies have been effectively developed to manage the delivery of growth factors for the purpose of enhancing bone regeneration”. The following issues could improve the scientific soundness of the review.

Major revision

1 Lane 72. The authors declared that the aim of this review was a critical overview on the efficacy of collagen membraned containing GFs developed by themself. The purpose seems too self-referential. In literature many papers discussed the use of collagene scaffold for GFs delivery in bone tissue regeneration, I propose to the authors to improved paragraph 4 with in-depth discussion in which their studies should be compared with those obtained in similar systems.

Minor revision

1 GBR and GTR in the incipit of introduction needed references.

2 The authors highlighted the crucial role of GFs delivery systems play in regenerative medicine (lines 62-65). In the introduction, I think would be appropriate to mention the use of autologus biomaterials such as PRF or CGF with a release kinetics of growth factor implicated in periodontal regeneration.

3 At the end of paragraph 2 the authors wrote “In addition, we look for an absorbable membrane that is suitable for bone regeneration units while investigating the toxicity of each membrane to tissues.” Could the authors explain this sentence? It would seem the aim of a research article.

4 Lane 299, there are two point next to reference n 72.

5 lane 341 “…and we are presented in Table1”. This statement seems wrong, could there be a mistake?

6 paragraph 5, lane 488. “The beneficial results obtained in this in vitro study are not yet sufficient to be used directly in clinical trials.” Which study are the authors referring to?

Author Response

We have made revisions in accordance with the Reviewers’ comments as follows:

Reviewer #1: The review discusses a very relevant area of regenerative science that is growing interest. The review reported the role of collagen membranes and several growth factor in bone regenerative medicine. The first part of the manuscript is well structured and discussed. Then the authors examine the use of CM with GF mentioning only their studies. However the authors themselves declare that “numerous controlled release systems utilizing membrane scaffolds including our studies have been effectively developed to manage the delivery of growth factors for the purpose of enhancing bone regeneration”. The following issues could improve the scientific soundness of the review.

Major revision

  1. Lane 72. The authors declared that the aim of this review was a critical overview on the efficacy of collagen membraned containing GFs developed by themself. The purpose seems too self-referential. In literature many papers discussed the use of collagen scaffold for GFs delivery in bone tissue regeneration, I propose to the authors to improved paragraph 4 with in-depth discussion in which their studies should be compared with those obtained in similar systems.

Response: We thank the reviewer for this suggestion. We added the following sentences and references in Introduction from lines 77 to 91. In addition, we edited the word from “emphasize” to “focus” in line 93.

Collagen-based materials, such as membranes, sponges, matrices, hydrogels, and composite scaffolds, are extensively utilized in in vivo studies to facilitate the regeneration of bone tissue[13]. The collagen sponge is highly regarded as a valuable biomaterial due to its exceptional functionality and properties, along with its convenient processing, sterilization, and storage capabilities[14]. Recently, Briquez et al[15] demonstrated a collagen sponge combined with bone morphogenetic protein (BMP)-2 and a bridge protein enhanced safety of BMP-2 and its beneficial impact on bone regeneration for spinal fusion. An effective strategy for bone regeneration was observed by incorporating a gene delivery system containing fibroblast growth factor (FGF-2) and BMP-2 within a collagen sponge, resulting in optimal bone regeneration effects[16]. Additionally, in the rabbit model of full-thickness bone defect regeneration, the utilization of a hydrogel collagen nanocomposite in conjunction with strontium and seeded mesenchymal stem cells improved bone formation[17]. The collagen/hydroxyapatite (HA) composite biomaterial scaffolds have been extensively investigated and used for bone tissue engineering scaffolds[18]. HA-collagen is very effective in healing critical-sized bone damage in a rodent model[19].

Minor revision

  1. GBR and GTR in the incipit of introduction needed references.

Response: We added the references.

  1. The authors highlighted the crucial role of GFs delivery systems play in regenerative medicine (lines 62-65). In the introduction, I think would be appropriate to mention the use of autologus biomaterials such as PRF or CGF with a release kinetics of growth factor implicated in periodontal regeneration.

Response: The reviewer had a great point and we agreed with it. We added the following sentences and references in Introduction from lines 66 to 72.

In recent years, autologous platelet concentrates (APCs), characterized by abundant GF levels, exhibit remarkable suitability for periodontal and bone tissue regeneration procedures. The main generations of APCs are platelet-rich plasma[10], platelet-rich fibrin[11] and concentrated growth factor[12] techniques. The advantage of the APCs is the fact that a large amount of GF can be delivered to the target site and encourage angiogenesis and wound healing.

  1. At the end of paragraph 2 the authors wrote “In addition, we look for an absorbable membrane that is suitable for bone regeneration units while investigating the toxicity of each membrane to tissues.” Could the authors explain this sentence? It would seem the aim of a research article.

Response: We have added the following sentence from lines 207 to 209.

It is necessary to have characteristics of biocompatibility, absence of biological side effects, efficient release of GFs, and biodegradability as the desired criteria for our study.

  1. Lane 299, there are two point next to reference n 72.

Response: We have deleted one.

  1. lane 341 “…and we are presented in Table1”. This statement seems wrong, could there be a mistake?

Response: We have deleted this sentence.

  1. paragraph 5, lane 488. “The beneficial results obtained in this in vitro study are not yet sufficient to be used directly in clinical trials.” Which study are the authors referring to?

Response: After we reconsidered it, we deleted the sentence.

Reviewer 2 Report

The manuscript "Growth factor delivery using a collagen membrane for bone tissue regeneration" written by Takayama and coworkers is an interesting review concerning guided bone regeneration.

Firstly, collagen membranes and their advantages/disadvantages are discussed in detail. The given overview is quite well but one very important topic is missing, namely that the nanotopography of collagen itself is already osteoinductive! If this 30 nm nanostructure is masked there is no bone regeneration at all. Please discuss and cite several relevant papers which are missing in your manuscript/references:

-Greiner et al, 2019 Nanomedicine, doi: 10.1016/j.nano.2019.01.018

-Vordemvenne et al, 2020 Cells, doi:10.3390/cells9030654

-Wähnert et al, 2021 Materials, doi: 10.3390ma14081961

The second part, GFs for osteogenesis, is well written and provide state of the art knowledge.

Overall the information in this review is of high interest for the scientific community and for clinicians.

Author Response

We have made revisions in accordance with the Reviewers’ comments as follows:

Reviewer #2: The manuscript "Growth factor delivery using a collagen membrane for bone tissue regeneration" written by Takayama and coworkers is an interesting review concerning guided bone regeneration.

Firstly, collagen membranes and their advantages/disadvantages are discussed in detail. The given overview is quite well but one very important topic is missing, namely that the nanotopography of collagen itself is already osteoinductive! If this 30 nm nanostructure is masked there is no bone regeneration at all. Please discuss and cite several relevant papers which are missing in your manuscript/references:

-Greiner et al, 2019 Nanomedicine, doi: 10.1016/j.nano.2019.01.018

-Vordemvenne et al, 2020 Cells, doi:10.3390/cells9030654

-Wähnert et al, 2021 Materials, doi: 10.3390ma14081961

 The second part, GFs for osteogenesis, is well written and provide state of the art knowledge.

Overall the information in this review is of high interest for the scientific community and for clinicians.

Response: We thank the reviewer for this suggestion. We added the following sentences and references in CMs section from lines 132 to 143.

The nanotopographical feature of collagen type I fibers directly stimulate the process of osteogenic differentiation in both human and mouse mesenchymal stem cells as well as neural crest-derived stem cells, without requiring additional biochemical signals for bone formation[25]. Moreover, it is widely recognized that nanotopography of artificial materials play a crucial role in determining the fate of stem cells[26], influencing gene expression, and promoting the attachment of bone-forming cells[27]. This nanopotography is shown to be solely sufficient for guiding osteogenic differentiation of osteogenic related cells in vitro[28]. In addition, transplantation of collagen sponge (Spongostan) with nantopographical structure into a critical-size calvarial rat bone defect further leads to fast regeneration of the lesion in vivo[29]. Therefore, it is suggested that the nanotopographical structure in the collagen membrane used in our study also played a crucial role in bone formation.

Round 2

Reviewer 1 Report

The authors have maintained the initial setting of the review, despite this the work has improved and I think it should be accepted.